# Experiences of Organisations of (or That Serve) Persons with Disabilities during the COVID-19 Pandemic and National Lockdown Period in South Africa

**DOI:** 10.3390/ijerph191912641

**Published:** 2022-10-03

**Authors:** Naomi Hlongwane, Lieketseng Ned, Emma McKinney, Vic McKinney, Leslie Swartz

**Affiliations:** 1Centre for Disability and Rehabilitation Studies, Department of Global Health, Faculty of Medicine & Health Sciences, Stellenbosch University, Cape Town 7602, South Africa; 2Interdisciplinary Centre for Sports Science and Development, Community and Health Sciences, University of the Western Cape, Cape Town 7535, South Africa; 3Department of Health and Rehabilitation Sciences, Faculty of Health Sciences, University of Cape Town, Cape Town 7700, South Africa; 4Department of Psychology, Faculty of Arts and Social Sciences, Stellenbosch University, Stellenbosch 7602, South Africa

**Keywords:** COVID-19, disability, health system, disabled people’s organisations, service delivery, South Africa

## Abstract

Organisations have long played an effective role in advocating for and actioning crucial developmental and humanitarian functions around the world, often under challenging conditions, as well as servicing the health needs of persons with disabilities. This article reports on the experiences of organisations of (or that serve) persons with disabilities, hereafter called service providers, during the COVID-19 lockdown period in South Africa beginning 26 March 2020. Organisations participated in an online survey as well as virtual narrative interviews to voice out their experiences. Five major themes emerged: (1) difficulties in keeping the doors open; (2) continued care under lockdown; (3) restructuring of care (4) government systems and policies; and (5) reaching out to offer and receive support. The findings demonstrate that the South African government failed to ensure targeted support to organisations of persons with disabilities. A remarkable feature of the organisations we interviewed for this small study was their agility in responding creatively to the challenges they faced, despite the difficulties. There is a need for government support to include targeted efforts to support organisation of persons with disabilities during pandemics to avoid worsening service gaps.

## 1. Introduction

People with disabilities experience inequities in health access and outcomes, and pandemics magnify these challenges. Evidence on perceived challenges and experiences of persons with disabilities are much needed to inform strengthening of disability-inclusive responses. South Africa reported its first case of COVID-19 on 5 March 2020, and on 26 March 2020, the South African government decided to impose nation-wide lockdown restrictions, which lasted until 5 April 2022 when the national state of disaster was lifted. Since then, South Africa has experienced four distinct COVID-19 pandemic waves [1].

There is a lack of data to understand how the pandemic affects people with disabilities (PwDs) who are more vulnerable to COVID-19 [2,3,4,5,6]. According to available evidence, PwDs are constantly exposed to significant risk and consequences as a result of pandemic crises, and they are now facing unjustifiable disadvantages in rationing critical care and life-saving treatment due to COVID-19 [5,6]. Prioritisation, also known as triage, is a source of concern for the disability community during disasters [7,8], particularly because, as we have seen in previous emergency situations such as Ebola and HIV/AIDS, people with disabilities are often left behind [9]. The lack of disability-inclusive disaster response and planning has exacerbated PwD’s existing structural disadvantages. Research into the impact of COVID-19 from the perspectives of service providers serving people with disabilities would be a good starting point for developing more effective and disability-inclusive systems. In light of this, we sought to empirically investigate the impact of COVID-19 and the associated countrywide lockdown restriction on organisations that represent (or serve) people with disabilities in South Africa during the first wave and to create evidence to assist future health emergency preparedness efforts.

### Background

Worldwide, the economy and health systems and/or services were disrupted and overwhelmed by the severe acute respiratory syndrome coronavirus 2 (SARS-CoV-2) outbreak, which significantly contributed to increasing rates of death. Declared as a virus which mostly affects vulnerable populations, the World Health Organisation (WHO) (2020) included persons with disabilities amongst those who are at risk of developing serious illness from COVID-19 and also more likely to die. This is well supported by evidence identifying that persons with disabilities are more likely to have underlying health conditions, coupled with discrimination and barriers to healthcare, information, social services, social inclusion, and education [10,11,12,13,14]. The Convention on the Rights of Persons with Disabilities (CRPD), an international human rights treaty, outlines people with disabilities’ fundamental human rights [15]. The CPRD has greatly informed the widely held belief that the rights of PwDs must be protected and promoted through both general and specifically designed laws, policies, and programs [16]. The COVID-19 pandemic has revealed that State Parties have not fully implemented the CRPD. It has starkly revealed the increased vulnerability and risks faced by people with disabilities because of entrenched discrimination and inequality. Furthermore, the outbreak of the COVID-19 pandemic in South Africa and the subsequent national lockdown have called into question national responses in the context of the right to the health of PWDs. As such, the WHO released a guiding document on disability considerations during the COVID-19 outbreak [14]. Similarly, the United Nations released a policy brief title “a disability-inclusive response to COVID-19”. Both these documents highlighted why additional disability inclusion is imperative during this pandemic. Additionally, these documents recommended actions that should be taken by governments across all sectors to ensure that persons with disabilities are not vulnerable. One such action is related to the need to undertake targeted measures for disability service providers in the community to ensure both access to and continuity of services.

South Africa has approximately 3.8 million PWDs, accounting for 7.5% of the total population. This excludes institutionalised people with disabilities and children under the age of five. The number of people with disabilities grows with age. More than half of those aged 85+ (53.2%) reported having a disability [17].

As within other lower-income countries (LMICs), persons with disabilities in South Africa are likely to have greater health needs, yet remain subjected to poorer access to health and social services [7,8,18,19,20]. Evidence shows these disparities were exacerbated by the COVID-19 pandemic [21,22]. It has also been shown that the demands of COVID-19 have overstretched an already pressured health system which has been failing to be disability inclusive [22]. This reality formed the basis for increased fears in the disability community about exclusion and its consequences.

Similar to other countries, South Africa declared its national state of disaster under Section 3 of the Disaster Management Act 57 of 2002 (Department of Co-operative Governance and Traditional Affairs 2020) and commenced its first hard lockdown on the 26 March 2020. The main purpose of this lockdown was to delay the spread of the virus and buy time for the health system to prepare its pandemic response. Some of the lockdown regulations included restrictions to population movement and face-to-face interactions to only immediate family. In addition, only essential services could be provided, and public transport was limited to certain peak hours and limited passengers. Under the regulations, healthcare services, supermarkets, and sole proprietors who provided essential goods and services such as spaza shops (also known as a tuck shop, is an informal convenience shop or small business in South Africa, usually run from home), as well as small-scale farmers, were deemed as essential services (Gazette 43258 2020). All other businesses wishing to provide essential services were required to seek approval from the Department of Trade, Industry and Competition (DTIC) in order to trade during the period of the lockdown (Gazette 43258 2020). With the economy essentially shutdown, the public experienced a combination of economic stressors, social isolation, job losses/insecurities, separation from loved ones, and health-related anxieties.

Within an emergency context, persons with disabilities are likely to rely on their organisations for various disability services and support. Many of these organisations play a crucial role as sources of care, support, and advocacy for persons with disabilities. There is evidence from the Disability Rights Monitor that, for decades, persons with disabilities and their representative organisations have effectively advocated for and performed crucial developmental and humanitarian functions on every continent and region of the world, often under challenging conditions [9]. However, during the hard lockdown in South Africa, a range of these services were either halted or restricted—a situation which contributed to disruption and a further decrease of available services [23,24]. Some organisations had to seriously consider alternative options, especially for persons with disabilities who have conditions that could potentially complicate if regular therapy or treatment was not provided. As such, more services were offered remotely during COVID-19 [22,25].

Emerging research at the time suggested that strict COVID-19 prevention measures were deepening pre-existing inequalities, with serious concerns about their effect on vulnerable populations, particularly PWDs [26]. In terms of healthcare service for instance, only patients seeking COVID-19-related healthcare could travel (Gazette 43258 2020). As a result, healthcare providers faced the difficulty of how to remain accessible to their clients without contravening these lockdown regulations [22]. In some situations, persons with disabilities were not allowed to be accompanied by their personal assistants [22].

Despite the recognised importance of equity and access in healthcare, the pandemic approaches and responses highlight continued access and equity challenges amongst persons with disabilities [21,27,28,29]. A survey study carried out by the Centre for Disability and Rehabilitation Studies [21] across South Africa revealed interruption of services needed, heightened fears and anxieties about themselves or their family members/caregivers contracting the virus, loss of livelihoods and income, and social isolation, amongst other issues. South Africa was also among the countries which provided the highest responses in the COVID-19 survey by the Disability Rights Monitor (83 out of 2152 responses from 134 countries) on human rights violations across 134 countries [9]. This high level of participation could signal an indication that serious concerns existed with regards to local pandemic responses and how these impact persons with disabilities [9]. Compounding the situation, disability remains absent in the disaggregated data collected on COVID-19 cases and deaths in South Africa—a structural case in point which serves to reinforce injustices because, as Kuper et al. [10] put it, what is not documented cannot be counted and factored into planning. This is despite calls by disability advocates and researchers to include disability in planning for effective health systems and disability-inclusive pandemic responses [10,30,31].

However, none of this is new and could be attributed to how disability is understood, as well as a history of systemic exclusion in an ableist society [21,22]. Such histories continue to shape service access and delivery. For example, the continued medicalisation of disability can be seen in how decisions within the health systems are shaped by medical ableist values [21,22,27]. This is evident in a study which revealed that only those with better prognosis tend to be favoured in accessing scarce resources such as assistive devices [32]. These decisions are often shaped by ableist values related to whose life is worth saving. The COVID-19 pandemic has similarly exposed this, as intense discussions about who will be prioritised for the already limited ventilator and intensive care units raised concerns in the disability community [27]. Savin and Guidry-Grimes stated that a balance is needed to consider when serving public interest and preserving healthcare systems so that, after a disaster, recovery remains possible [33].

Against this background, we aimed to describe the experiences of persons with disabilities and organisations of (or that serve) persons with disabilities during the COVID-19 pandemic’s national hard lockdown period in South Africa as well as identify possible strategies and recommendations for disability-inclusive pandemic responses. This paper reports only on the experiences of organisations.

## 2. Materials and Methods

This was a mixed-method descriptive study which collected experiences of persons with disabilities and their organisations across the nine provinces of South Africa. However, responses were received from eight provinces. This paper focuses specifically on the data from the participating organisations. This study was concerned with the first two hard lockdowns that took place in South Africa. The lockdown had different five stages which varied according to intensity of restrictions imposed. The first intense lockdown (level 5) took place from 27 March 2020 until 30 April 2020 when a risk adjusted phase which included a phased re-opening of the economy was introduced. This level indicated high spread of COVID-19 with low readiness of the health system. Level four was introduced from 1 May 2020 to 31 May 2020. This level indicated some moderate to high spread with a low to moderate health system readiness.

We collected data through an online survey (using the SUNSurveys platform (Stellenbosch University, Cape Town, South Africa)) which was distributed through the existing disabled people’s organisations (DPOs) and other disability networks country-wide using emails and social media platforms. DPOs in South Africa are membership-based organisations which are managed and controlled by persons with disabilities. They constitute the representative voice of PWDs on the basis of their demographics. Organisations run by parents of children with disabilities also fall under this classification of DPOs in South Africa. Of the 39 national DPO’s available in SA as listed in the Ministry of Women, Youth and Persons with Disabilities, a total of 11 organisations participated on the survey. The African Network of Evidence to Action on Disability is one key network which has a national reach to all organisation in South Africa, and this network also distributed the survey. While some of these organisations have a service presence in all provinces of South Africa, others only work in one or two of the provinces. The surveys were only presented in easy-read English (as budget limitations precluded translation to all South African languages) but was compatible with disability-related software used by people with visual impairments (such as screen-readers and font-size adjusters). Before distribution, the survey was piloted in two rounds, and feedback was used to improve the survey. Two members of the research team identify as persons with disabilities and assisted with improving the survey.

The survey included questions under six thematic areas, namely, about the organisation; access to information on COVID-19; access to health facilities and services; access to other services; and participation in decision making and governmental involvement. We drew on these thematic areas from reviewed literature as well as key documents, such as the United Nations Report [26] on considerations for a disability-inclusive response to COVID-19, as well as the World Health Organisation’s 2020 report [14] on disability considerations during the COVID-19 outbreak. During the distribution, the organisations were given a brief intended purpose of the survey and were invited to participate and circulate the link to their membership. It was also specifically mentioned who is eligible to participate in each survey, and contact details for queries were provided.

On the basis of our survey, any employee or representative of the organisation could complete the online survey. Seven of these organisations were represented (predominantly) by the director of the organisation. The roles of other respondents were a caregiver in the organisation (one); a rehabilitation therapist (one); and national programs managers and coordinators (two). Nine of these participating organisations provided services to a range of all disability types, one addressed only physical impairments, and one addressed all impairments except intellectual disabilities. The services provided in these organisations covered educational support; advocacy and training; economic and skills development; special care (residential and day care services); assistance with healthcare of persons with disabilities, including counselling; and support groups and capacitation of persons with disabilities and their families.

At the end of the survey, we asked the respondents to indicate if they would like to participate in follow-up narrative interviews. Only six respondents could be reached for these interviews. The main offices of these organisations are located across three provinces, namely, Gauteng Province (GP), Free State (FS), and Western Cape (WC). The service providers provide services for children and adults with disabilities, including those with epilepsy and physical, learning, and/or multiple disabilities (E = epilepsy, CWD = children with disabilities, PWD = people (children and adults) with disabilities).

We conducted in-depth follow-up interviews virtually via a format convenient for the respondent including Zoom, MSTeams, and by request WhatsApp. These data are from the first wave of the pandemic. Ethics approval was provided by the Research Ethics Committee of Stellenbosch University (REC-2020-15244). Participants were assured of anonymity, and their real names were not used in reporting.

There was one interview per person, as this was a follow-up from the survey responses. These lasted between 30 and 60 min and were transcribed verbatim by research assistants. Two researchers (the PI and co-investigator) conducted the first level of analysis. The other two co-investigators cross-checked the analysis against raw data. Data from the survey were descriptively analysed using the mentioned above six key thematic areas of the survey. For interview data, we used the six stages of thematic analysis by Braun and Clarke as the framework for coding and identification of themes [34]. That is, after repeatedly reading the transcripts of the interviews (by two authors) for immersion and to become familiar with data collected, we started generating codes from raw data using the colour coding framework. Data coders consulted with each other regularly during the process to confirm emerging codes from data. These codes were refined and clustered into possible overarching themes which were discussed with and reviewed by other team members and the interviewers. These themes were further synthesised, finalised by incorporating feedback received, and grouped according to the International Classification Specifications of Functioning, Disability and Health (ICF) (WHO, 2001) environmental factors. Lastly, we produced a combined report. In this paper, we focused specifically on themes related to services, systems, and policies as one aspect of environmental factors.

## 3. Results

The findings on this paper include survey responses of 11 organisations of people with disabilities and subsequent follow-up interview data from six organisations. The results focus on five major emergent themes: (1) difficulties in keeping the doors open; (2) continued care under lockdown; (3) restructuring of care; (4) government systems and policies; and (5) reaching out to offer and receive support. Table 1 is a thematic representation of the themes that emerged. These themes describe the experiences, perspectives of organisations on challenges and facilitators faced during the stated period particularly relating to systems, services and policies in place, and how these shaped their experiences.

### 3.1. Difficulties in Keeping the Doors Open

Service providers expressed difficulties in keeping their organisations opened. The following two themes emerged from the data.

#### 3.1.1. Closure of Services

Service providers found it difficult from a governance and management point of view, especially when deciding what services to close or keep open, and closure had a great impact on the provision of basic services.

Some service providers described that they had to close most, if not all, of their services when lockdown was enforced:


*“several of our homes in KZN and Eastern Cape also have day care centres for children with physical disabilities services had to be closed down since the beginning of lock down, that portion of our services had to close down altogether, those children have had to go back to their families that has affected hundreds of children with disabilities”.*
(GP, CWD)


*“After the President’s address to close schools we completely stopped”.*
(GP, PWD)


*“That portion of our services [day care centres for children with physical disabilities] had to close down altogether”.*
(FS, PWD)

The closure led to a disruption of services. To some organisations, this meant that social workers could no longer be sent into the communities for face-to-face home visits or community outreach programs, thus depriving people of much-needed services:


*“… we cannot go to communities during the lockdown”.*
(WC, E)

Counselling services in hospitals also had to stop and residential homes could not carry on with their normal activities. Furthermore, services providing care and education for children with disabilities were brought to a halt to prevent any potential risks.

Besides education, children with disabilities also received food on a regular basis when they were at the day centres and service providers expressed concern on how children would access basic needs and services during closure:


*“so when the centre is closed there is no place where they get food”.*
(GP, PWD)

Furthermore, they feared how children at home would be looked after while their families tried to access basic necessities:


*“The caregiver must leave home in order to access food relief or medical access and kids are left unsupervised”.*
(WC, E)

Service providers shared that their organisations relied heavily on donor funds and other fund-raising initiatives. However, these activities were brought to halt as a result of lockdown regulations:


*“Normally our fundraising is in the form of events and because of social distancing and the lockdown we can’t do any of that”.*
(FS, PWD)

Service providers were gravely concerned about the lack of ongoing therapy and educational services for vulnerable children and their families, especially as children missed out on their otherwise daily access to early childhood development (ECD) centres during lockdown. Educational stimulation programmes which had greatly benefited the development and understanding of children with impairments, including those physical disabilities, cerebral palsy, and autism, could not reach these children who had to stay at home and service providers were fearful that the children’s development would be halted or regress considerably:


*“the stimulation programmes are run as if we are in a main stream schools, lessons are offered in every subject”.*
(GP, PWD)

Closing of day centres also had a significantly negative impact on families who now had to look after their children with disabilities on a constant basis.

#### 3.1.2. Losing Wages and Income

The economic downturn resulting from the pandemic had a substantial impact on (un)employment within organisations. Many staff members had to rely on unemployment insurance funds (where available) for a number of months while service providers could not pay their salaries. A number of staff had disabilities (35 members in one organisation) and had to be laid off completely, thereby losing income and becoming reliant on a disability grant (WC, E).


*“Loss of income for staff and employees with disabilities in the protective workshop”.*
(anonymous survey participant)


*“Risk of resignations as professional staff (social workers) seek alternative employment”.*
(anonymous survey participant)

### 3.2. Continued Care under Lockdown Restrictions

Some services managed to stay open; however, they encountered challenges as they struggled to maintain daily operations as resources dwindled in pursuit of continuing with care. Service providers described how they were left with no choice but to keep operating:


*“we have residential homes for people with physical disabilities, and because they are residential homes those homes the services has not been put on hold, we could not send all our residents back to their families, the main services which is accommodation and care service have been continuing uninterrupted”.*
(FS, PWD)

Having to feed residents was singled out as a notably difficult burden; however, some managed to arrange financial and food support for children with disabilities and their families, as well as ECD centres.


*“Also for ECDs in Phillipi [a township in Western Cape], we carried on with feeding scheme even though school has been closed”.*
(WC, CWD)

#### 3.2.1. Incorporating Protective Measures in Care

Service providers did their best to implement strict protection measures to protect their care workers as well as their clients with disabilities. However, given the hands-on nature of caregiving, especially within the residential homes, implementation of social distancing was not possible between caregivers and residents. Given the vulnerability of residents with disabilities, caregivers made sure they sanitised their hands regularly, and social distancing between residents was encouraged. The service providers place great value on the safety of their staff in order to continue to provide services and expressed a strong sense of responsibility towards maintaining a safe workspace for them. Caregivers at residential homes were provided with the appropriate masks, gloves, and aprons, and followed stringent protocols regarding hygiene and inflection control as much as possible (WC, CWD) (FS, PWD).


*“We have placed emphases on the use of PPEs”.*
(FS, PWD)

However, in the attempt to continue with care many service providers lacked sufficient (or the appropriate) protective equipment that they needed to implement these measures for their staff so they could carry on with their programmes:


*“Sometimes we are short of mask, sanitizers, aprons, etc.”.*
(GP, PWD)


*“If we can get such things when we going to attend the kids, then we know we have safety gear all ready”.*
(GP, PWD)

Besides lacking resources, service providers also struggled with messaging on issues of social distancing and other protection measures.


*“Educating blind people on social distancing and sanitizing—getting extra sanitizers for the PAs—re-educating the Deaf on social distancing. COVID-19 rules are contrary to their cultural norms—knowing (not knowing) which sanitisers are not harmful to those with albinism”.*
(FS, PWD)

Most service providers relied on donations for protective equipment, and some received assistance from the Department from Health for provision of PPE for COVID-19.

#### 3.2.2. Impact of Fake News

Service providers indicated that fake news disrupted the continuation of services, and they had to take measures to try and prevent them.


*“Because that [fake news] increases stress. But is something that sometimes you cannot manage”.*
(WC, E)


*“Like the [fake] news spread that somewhere they are giving away food, we had to advise our beneficiaries to be careful not to believe in everything they hear”.*
(WC, CWD)

Some organisations collaborated with the Department of Health to address the issue of fake news, while others had not been affected by it at all.

### 3.3. Restructuring of Care

To counteract the impact of lockdown, many service providers were re-examining their operations and conducting surveys with their clients towards re-structuring how they offered care, looking at ways to become more inclusive, and maintaining relevance and sustainability. The use of technology, social media platforms, and adapting conventional ways of caring were key in restructuring care in a pandemic era.

#### 3.3.1. Adapting Conventional Ways

Adaptations included becoming a remote centre of resource and support instead of physically visiting clients. Initiatives extended to delivery of stationery and educational resources, such as educational simulation materials, via community workers to principles of ECD centres as well as children with disabilities and parents at home (WC, CWD). Some service providers introduced 0800 numbers that persons with disabilities and/or their families could contact for further information.


*“What we doing now our social workers are calling clients, just to find out what the situation are at the moment”.*
(WC, E)

Service providers described how they expanded their existing health forums, generally used for communicating with forum members, in order to disseminate and spread news throughout the rest of the community (WC, E).

#### 3.3.2. Technological Innovations: Tools for Continued Care

The use of technology, particularly cellular phones, the Internet, and social media platforms, had a significantly positive impact on the ability of organisations to provide their services remotely. Service providers also made regular use of e-mail and social media platforms such as WhatsApp and Facebook to maintain communication with clients and staff (WC, CWD), disseminate information, and create support groups and forums.


*“I created a WhatsApp group for all the employees with disabilities, so that they could be updated”.*
(WC, E)

WhatsApp was also used effectively for face-to-face therapy sessions. Service providers and parents alike made use of video to assess and provide feedback on the development of babies with disabilities.


*“We look at the video [of the baby being active] together as speech; physio and occupational therapist come up with the home programme, and send them back to them via WhatsApp, messages, in writing or in video clips”.*
(WC, CWD)

However, despite these initiatives, recipients of care were faced with challenges in using social media, as issues of affordability and access were echoed. Making use of video could be expensive, and often parents had to be provided with data (WC, CWD).

It was highlighted that not everybody, especially those who were most vulnerable, could afford to have smart/cell phones and/or connect to the Internet or media platforms such as WhatsApp. In many of these cases, regular telephone calls or SMSs were used to maintain contact. However, some people were disabled to the extent that they could not access any communication technology or social media (FS, PWD), highlighting their lack of access information and isolation.

### 3.4. Government Systems and Policies

#### 3.4.1. Inadequate Government Regulations and Guidelines

Most service providers noted that the government guidelines on COVID-19 did not accommodate disability and were considerably inadequate, unclear, and sometimes contradictory regarding what procedures to follow in an emergency disability response, such as social distancing and wearing masks within a care environment, and this affected how they offered care. This lack of clarity created unhappiness amongst many residents, staff, and management within residential homes:


*“[There is] confusion with regards to how much/long must one isolate, can our residents go out, can they not go out…how to manage of staff without infringing on the staff rights to health”.*
(FS, PWD)

Conflicting legislation around labour and COVID-19 rules and regulations made it difficult for service provider management, creating the potential for caregivers to go on strike and refusing to care for residents.


*“There have been issues amongst staff in terms of are they going to still be required to come to work if any one of the residents do test positive”.*
(FS, PWD)

Service providers also identified the need for options to make information accessible, including sign language interpreters working with community organisations and persons with disabilities,


*“… on how information is shared and how people can access services”.*
(WC, CWD)

Some service providers remained unsure about the rationing system regarding who should give access to ventilators/intensive-care treatment, but did highlight that the government, and its legislation, were generally unprepared for COVID-19:


*“It [lack of clear government guidelines] gives me a certain amount of anxiety”.*
(FS, PWD)

#### 3.4.2. Government Relief Aid

Facing depletion of their relief funds, many service providers applied for government funding as soon as they could. It was felt, however, that the government did not provide nearly enough funding for the disability sector and that their structures were not equipped to handle funding application processes adequately:


*“[They] clearly didn’t have the resources to process the applications that they received”.*
(FS, PWD)


*“That 150 mill for all the NPOs in the country is nothing …it was not enough and have taken far, far too long”.*
(FS, PWD)

Processes implemented for verification purposes, put in place to minimise corruption, only made access to aid a prolonged and tedious exercise:


*“Other agencies’ information, to check if everybody is legitimate, so you have to fill in many forms, that requires ID numbers”.*
(WC, E)

Even those who followed due processes fell short and were left disappointed:


*“we think the government is trying to help but the help doesn’t reach everybody because since we applied for the food parcels, we applied for nappies nothing has ever reached us up to this far”.*
(GP, PWD)

For PWDs, as well as many parents of children with disabilities, received a disability grant as their only income (WC, E). During lockdown, the government increased the disability grant by ZAR 350, initially for a period of six months, and this extra amount was a big help:


*“I can tell that additional SASSA grant [helps]”.*
(FS, PWD)

Service providers hoped that the grant increase would be extended for longer (FS, PWD).

The majority of persons with disabilities received their disability grant directly (WC, CWD). However, those who lived in residential homes paid about 75% to the home and received the rest for their own pocket money, for which they were grateful (FS, PWD).

Service providers were quick to highlight that disability organisations were not consulted regarding allocation of government funds and felt excluded by the process. They also identified that many held the misconception that services for the disabled were more expensive (WC, CWD):


*“We feel as if we have been forgotten, we feel as if we are an afterthought”.*
(FS, PWD)

### 3.5. Reaching out to Offer and Receive Support

The data revealed the challenges of providing support within the community, support within the organisation, and obtaining support from outside the organisation. This is represented below by the following themes.

#### 3.5.1. Stepping in to Provide Support for Stopped Government Programs

Concerned for the ongoing development of babies with disabilities, a service provider stepped in to assist parents by carrying on with government programs that had ceased:


*“We even taken back some of the babies whom were discharged from the government therapy service because those were stopped”.*
(WC, CWD)

Some service providers also intervened financially to help families where government assistance was lacking:


*“We were able to give some of the families monetary amount to help themselves with grocery, toiletries etc”.*
(WC, CWD)

#### 3.5.2. Remote Support and Distributing Resources

Service providers were inundated with calls from members of the community and found it extremely difficult to structure community support during lockdown (GP, PWD) (WC, CWD) (FS, PWD) (WC, E). They continued, however, to reach out and provided important remote support where they could (WC, CWD). Some also provided hands-on support through activities such as distributing resources between schools and home, delivering medication, and helping families who have children with severe disabilities:


*“We have been trying to support the kids, like by going there, especially the ones who cannot bath themselves”.*
(GP, PWD)

These caregivers were mindful of following stringent protection measures when entering homes:


*“This [wearing masks and seeing children with disabilities at their homes] will make parents trust us and for our safety”.*
(GP, PWD)

The data also revealed the importance of direct psychosocial and emotional support during lockdown for persons with disabilities, their families, and ECD centres/schools (WC, CWD). Service providers provided emotional support and shared information with schools and parents. They also tried to intervene before situations became abusive, supplying hotline numbers and other resources for those who needed it.

#### 3.5.3. External Support

The lockdown situation highlighted the need for government, community organisations, and persons with disabilities to develop stronger partnerships and support each other more (WC, E) (GP, PWD).

Service providers recognised the value of their own community knowledge and the need to enhance this for future collaboration:


*“Because the NPOs are specialists, we know what’s going on in the communities and we know our clients, we will be able to render services”.*
(WC, E)

The lockdown situation also gave rise to new partnerships with service provider contacting other districts for support with delivering food parcels:


*“At least it was a break through, in that type of communication”.*
(WC, E)

Others benefited from good relationships and support from their regular donors:


*“We are literally sent out a message to our supporters and said we need money, and our supporters have been extremely generous”.*
(FS, PWD)


*“We have a good relation with [pharmaceutical chain] and they have been very helpful with all our homes”.*
(FS, PWD)

Some service providers received assistance from the state (FS, PWD), but many were still waiting for support:


*“We think the government is trying to help but the help doesn’t reach everybody”.*
(GP, PWD)

## 4. Discussion

This study explored the experiences of organisations of (or that serve) persons with disabilities. The findings strongly indicate that many disability services were interrupted. They also showcase that civil society and community organisation of persons with disabilities lacked substantial support from the government and that there was no significant consultation from the government to ensure inclusion in the design of rapid disability inclusive measures. As such, participating organisations detailed the limited support they received as well as the implications on their service provision. Despite the disability legislature and frameworks available in South Africa as well as the strong emphasis on disability rights as promulgated in various constitution sections, it is clear that the framework and approach to responding to the pandemic still lacked a disability focus. While persons with disabilities are said to be included in “vulnerable groups”, these findings expose how their needs and support were not factored in the responses aimed at protecting those who are more vulnerable to the pandemic. These perceived challenges will inform key actions needed for improving accessible and inclusive systems during a pandemic as well as avoiding interruption of services which put people at higher risk and more vulnerability.

Many agencies had predicted that COVID-19 will disproportionately affect persons with disabilities, putting them at risk of increased morbidity and mortality [9,10,11,21]. As such, governments across the world were advised to design varying strategies to support disability inclusion in formal responses to the pandemic [14,26]. However, governments around the world have been slow to recognise the unique and diverse health needs of children and adults with disabilities during the COVID-19 pandemic [35]. This study revealed that the COVID-19 pandemic and the restrictions implemented have had a differential impact on this disadvantaged group and exacerbated pre-existing challenges of access to quality healthcare embedded within the South African healthcare system. Shakespeare et al. [3] echoes this notion and mentions three factors that could be reasons for this differential impact, namely, that persons with disabilities are at an increased risk of poorer outcomes from COVID-19, they have reduced access to routine healthcare, and that there are adverse social impacts of efforts to curb this pandemic [22]. Indeed, our study confirmed these factors and shows how the disadvantage was compounded.

In light of these factors, this study also highlights that COVID-19 has negatively impacted how organisations deliver services to persons with disabilities. McKinney and colleagues state that disability-specific health services were not considered as essential services in the initial stages of lockdown in South Africa [29]. This is despite the fact that these organisations also play an imperative role, support the Ministry of Health in delivering health and other services to the disability community. This de-prioritisation was evident, as some organisations in the study had to close their doors, lose access to essential services, and had their employees with disabilities facing job losses. Those offering educational programs and stimulation groups to children with disabilities shared their worry of these children regressing and deteriorating. Prior studies have shown that due to the inadequate preparedness for the aftermath of COVID-19 on persons with disabilities, education provision has been overlooked and was not considered for children with special needs [36,37,38]. Expanding community services has been highlighted as a key way of protecting persons with disabilities as growth in demand for support services in the community without a corresponding supply of services can increase pressure on families [26]. Given that we did not have this preparation in the country, families and persons with disabilities were mostly left unsupported [21,22]. Furthermore, those who continued with care because they housed persons with disabilities had to manage the infiltration of fake news that created fear and anxiety among staff and residents, all the while being burdened with implementing COVID-19 protective measures that were impractical for some persons with disabilities as per their respective impairment needs.

In a scoping review conducted by Kamalakannan et al. [6], it was found that environments within long term or residential facilities created a “perfect storm” regarding susceptibility to the pandemic, especially those for residents with disabilities because of the shared use of space and high level of personal care assistance from staff [6]. Organisations expressed that they tried to enforce all non-pharmaceutical measures; however, explaining social distancing to the deaf and blind was a challenge, and dwindling PPE resources presented an additional barrier to the implementation of these safety measures. A similar finding from a different study reported staff working in multiple care homes who experienced lack of training and inadequate personal protective equipment (PPE), which were described as factors that increased the risk of transmission [39]. It is worth mentioning that these happen against the backdrop of the recommendation to governments to ensure that disability organisations access PPE at no cost [14,26].

The study further revealed that organisations that continued operating had to be innovative and adapt their usual ways of rendering care, and this included the use of technology and social media platforms. Social media has played a central role in online interactions and has been shown to have played an important role during the COVID-19 pandemic in particular [40]. However, this study also indicated that despite online technology being used as a tool to deliver care, it was not without challenges. Considering that the majority of the recipients of care from the organisations in this study were from poor backgrounds living in rural areas issues of network connectivity, access to smart phones or computers and funds to purchase data were barriers to accessing care. This makes the innovation of telemedicine a luxury for a selected few.

The study highlighted that the response from government systems and policies was limited because of a lack of targeted planning and prioritisation of organisations of persons with disabilities. As such, some organisations sought external support when government aid and distribution of resources between organisations and persons with disabilities did not suffice. In the backdrop of pre-existing inequalities faced by persons with disabilities, the impact that COVID-19 made in exacerbating these inequalities was revealed in the study. International organisations such as the World Health Organisation (WHO) and United Nations (UN) tabled policy guidelines that aimed to respond to such challenges and foster disability inclusion in the global response to COVID-19. The World Health Organisation (WHO) established planning guidelines in support of countries’ preparation and response to COVID-19, and of particular interest to persons with disabilities is the emphasis on cross-sectoral partnerships and collaborations. Key guidelines include ensuring that public health information and communication is accessible not only in its reach but in the format it is presented in, in order to ensure meaningful involvement and active participation of persons with disabilities and their representative organisations in all stages of the COVID-19 response and undertaking measures that ensure there is continuity of care [14,26].

Another recommendation was for governments to ensure access to short-term financial support for disability services to ensure they remain financially sustainable if they experience a downturn [14]. Organisations in this study shared that there was a distinct lack of targeted efforts towards disability organisations from the government, and furthermore, messaging on regulations were at times unclear. The South African government has seemingly failed to effectively support disabled people’s organisations (DPOs) through its emergency social protection measures, as this study provides further evidence of the dire situation they faced during hard lockdowns and after such lockdowns. For instance, there were no targeted efforts to support DPOs which run on the basis of government funding to improve their access to government support measures such as the funding which was availed for small businesses in general. This is unsurprising, given that the Disaster Management Act 57 of 2002 for the country also excludes persons with disabilities and any disability-inclusive approaches. This demonstrates how government policymakers continue to fail in recognising the importance of user-led and disabled people’s organisations even in planning and policy making (see key recommendation in Box 1 below). This is also indicative of continuous lack of disability inclusiveness and accommodations, despite the existence of disability policies in the country which cover various aspects of disability rights.

Box 1Key recommendations for policymakers.
-The government must prioritise supporting organisations of people with disabilities to ensure uninterrupted access to services at community levels and prioritise access to protective equipment.-Disability inclusion must be at the centre of all stages of COVID-19 response and recovery to ensure inclusive pandemic responses.-Input from people with disabilities and their representatives, through collaborations and partnerships, must be considered when designing measures, as this will ensure effectiveness and accountability.-There is a need to have targeted measures for such support networks/disability service providers such as considering short-term financial support for disability services.-The Disaster Management Act must specifically include people with disabilities.-It must be ensured that disability services are considered as essential work and exempted from curfews and other lockdown measures which affect service provision.


At an individual level, one of the responses to the COVID-19 crisis and regulations was to provide relief to vulnerable individuals and households including persons with disabilities. In this light, the South African government expanded its unconditional social grants on both the intensive (an increase in the amounts of all existing transfers) and extensive (the introduction of a special COVID-19 Social Relief of Distress grant) for six months from May to October 2020. Additionally, more funds were released to distribute food packs. However, for the most part, organisations that support persons with disabilities here were struggling to function or some closed due to regulations and thereafter financial implications of COVID-19. It is likely that this situation created further access gaps to resources such as food packs, information, and other health services rendered by these community organisations, especially considering that this group has been facing access gaps to social protection even before COVID-19. This is why the COVID-19 DRM calls on duty-bearers and signatories of the CRPD to remember their commitment to the inclusion of persons with disabilities and their representative organisations in any and all government responses to COVID-19. As they put it, there can be no inclusive response to COVID-19 without the involvement of persons with disabilities and their representative organisations [9].

### Limitations

We note that a common feature of surveys is that of more likely getting those with complaints to respond than others—a feature which introduces a biased sample. Due to limited face-to-face research, as per the research ethics regulations for COVID-19, we could not change our methods of data collection to enable more participation from those without access to electronic devices and internet connectivity. Similarly, those living in remote settings were likely excluded due to poor infrastructure and connectivity issues. While a significant increase in the use of virtual methods in data collection is noted, challenges related to access to data, smartphones (and similar communication devices), and network connectivity still exists in South Africa. This limited broader participation. An additional issue is that there were many surveys rolled out during the first wave and this could have resulted in research fatigue in many. Furthermore, the survey was only available in English and could not be translated into other South African languages due to limited funding. Other groups which could have been hindered from participating are people in residential care homes or institutions. We are therefore aware that the study may have been less accessible and inclusive than a face-to-face process would have been particularly regarding the diversity of respondents. However, despite these limitations, we believe that the findings strongly indicate the challenges experienced by many people with disabilities in our context.

## 5. Conclusions

A remarkable feature of the organisations we interviewed for this small study was their agility in responding creatively to the challenges they faced, despite the difficulties. It may well be the case that the small sample who responded to our request for participation in the study were better-functioning and better-resourced organisations; the situation for less well-resourced organisations is likely to be worse. An important contribution to the debates on services under the pandemic from the disability organisations is not only that these organisations highlight challenges faced by a particular vulnerable group—persons with disabilities. In addition to this, the challenges faced by these service organisations provide a way in to begin to understand broader concerns about access in a deeply unequal society, one in which services which are taken for granted in better-resourced sectors and contexts cannot be assumed to be present. In the context of a pandemic, for example, the issue of lack of access to medical supplies affects all people, and the issue of digital access (and, indeed, digital justice), although clearly crucial for persons with disabilities, is also of great relevance to many others in the population at large. As South Africa and other countries face ongoing challenges and restrictions into the future, the experiences of the organisations we discuss here, and their ways of dealing with them, have continued relevance.

## Figures and Tables

**Table 1 ijerph-19-12641-t001:** Representation of themes.

Themes	Sub-Themes
**Difficulties in keeping doors open**	Closure of services
	Losing wages and income
**Continued care under lockdown restrictions**	Incorporating protective measures in care
	Impact of fake news
**Restructuring of care**	Adapting conventional ways
	Technological innovation: Tools for continued care
**Government systems and policies**	Inadequate government regulations and guidelines
	Government relief aid
**Reaching out to offer and receive support**	Stepping in to provide support for terminated government programs
	Remote support and distributing resources
	External support

## Data Availability

A dataset will be submitted upon request by the editor.

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
