# Peer review of "Experiences of Organisations of (or That Serve) Persons with Disabilities during the COVID-19 Pandemic and National Lockdown Period in South Africa"

_ijerph, 2022, doi:10.3390/ijerph191912641_

Round 1
Reviewer 1 Report
Thank you for the opportunity to review the manuscript, 'Experiences of organizations of (or that service) persons with disabilities during the COVID-19 pandemic and national lockdown period in South Africa'. The paper requires major revisions, and I sincerely hope that the feedback to the authors below can assist in their efforts to revise and strengthen the scientific rigour and compelling nature of the paper. In making these recommendations, I confirm that I have no professional or personal relationship currently or previously with any of the authors on the paper.
1. Introduction
The final sentence of the 7th and final paragraph of the Introduction set out what the purpose of this paper/study is. By the time I got to the 7th paragraph, this was somewhat too late. I would strongly recommend that the authors rework the Introduction by providing a strong introductory paragraph, which is only 1-2 paragraphs maximum, outlining the purpose of the paper and why this is an important and novel contribution to the literature, not only for a South African audience but for the International Journal of Environmental Research and Public Health's international audience. Much of the remaining 7 paragraphs could then be Background.
The authors do not clarify what is the COVID-19 lockdown period under examination in the study. They state in para 1 (intro) that is begins March 26, 2020 - but until when? They indicate the study occurs during the first wave of the pandemic in South Africa, but there really needs to be greater clarification of the study period/timeline.
The definition of 'organisations' also needs to be clarified up front - are the authors referring to government, civil society, faith-based, private sector??
Please ensure that all statements of fact are referenced. For example, please reference 3rd sentence of para 1 (intro) "Declared as a virus which mostly affects vulnerable populations..."; final sentence of para 4 (intro) "As such, more services were offered remotely during COVID-19" (are you referring to telehealth? please reference).
Para 2 (intro) the figure that 3.8 million people in South Africa who experience disability needs to be contextualised against the figure of the country's overall population - that figure was not given.
Para 2 (intro) 'As within other contexts,' - do you mean country contexts?? Why is there no mention of the CRPD and concerns about the impacts of exclusion on the rights of persons with disabilities in South Africa, i.e. South Africa's obligations under the CRPD??
Para 3 (intro) - please clarify what 'spaza' shops are for your international readership.
Para 4 (intro) - where possible, please avoid the phrase 'disabled people' (i.e. sentence 1). This language does not comply with the CRPD.
2. Materials & Methods
I have several concerns about the scientific rigour of the research as presented in the Methods section. As I have concerns about the Methods, I therefore have concerns about the validity of the research findings and the appropriateness of the Conclusion(s) based on the Results. I'd strongly recommend the authors closely rework the Methods section. Methods is/are everything.
- please clarify the time period under examination in this study; what is the national lockdown period in South Africa that is the period of focus in this paper?
-I would like to please see the online survey attached in a Figure in this paper. What were the questions asked?
- How many organisations were originally contacted to participate in this study - 11 agreed to participate, but how many were originally contacted to participate?
-What type of organisations participated in the study - I would like to please see a Box containing this information i.e. (1) how many were civil society, government, faith-based, private etc? (2) What were the general disability service provision focus of each of these organisations?
-Did the interview participants receive some sort of Acknowledgement for their time? How was their time/engagement in this study Acknowledged? How was consent obtained?
- Why were people with disabilities not part of the cluster of interview participants/findings presented in this study/paper? Will the authors present those results of the voices/lived experiences of persons with disabilities elsewhere? Have they already done this??
- In the final para of the Methods, the authors inform that the qualitative data was subject to thematic analysis? Who/which literature guided and grounded the research team's approach to the thematic analysis? How was the data thematically analysed and synthesised with the survey results - what was the rigorous process that engaged, who was involved? How can I as the reviewer be assured of research rigour and robustness, inter-rata reliability etc?? 'We generated codes using different colours' needs to be removed/reworked? This is not appropriate 'scientific' language. Please revise.
-para 4 (Methods) ('Due to limited face-to-face research...') describes study limitations, which should be moved into the Discussion section of the paper.
3. Results
- As mentioned, because I have concerns about the Methods, I have concerns about the validity of the Results. The fact there is only 3 key themes, with many subthemes, compounds these concerns. I would suggest the authors could present 5-6 themes. Please review the raw qualitative data findings.
-The first sentence of the Results needs to be much stronger - please have a look at other qualitative research papers for guidance.
4. Discussion
For the Discussion to have impact, please rework the initial paragraph (again please summarise the findings).
Please summarise the key recommendations in a Box for policymakers - this is important for increasing the IMPACT of your research.
Author Response
Dear reviewer please see attachment.

Reviewer 2 Report
Please see the attachment.

Author Response
Dear Reviewer
Please see attachment.
